# The Alterations and Roles of Glycosaminoglycans in Human Diseases

**DOI:** 10.3390/polym14225014

**Published:** 2022-11-18

**Authors:** Qingchi Wang, Lianli Chi

**Affiliations:** National Glycoengineering Research Center, Shandong University, Qingdao 266237, China

**Keywords:** glycosaminoglycan, human disease, ECM remodeling, heparan sulfate, chondroitin sulfate, dermatan sulfate

## Abstract

Glycosaminoglycans (GAGs) are a heterogeneous family of linear polysaccharides which are composed of a repeating disaccharide unit. They are also linked to core proteins to form proteoglycans (PGs). GAGs/PGs are major components of the cell surface and the extracellular matrix (ECM), and they display critical roles in development, normal function, and damage response in the body. Some properties (such as expression quantity, molecular weight, and sulfation pattern) of GAGs may be altered under pathological conditions. Due to the close connection between these properties and the function of GAGs/PGs, the alterations are often associated with enormous changes in the physiological/pathological status of cells and organs. Therefore, these GAGs/PGs may serve as marker molecules of disease. This review aimed to investigate the structural alterations and roles of GAGs/PGs in a range of diseases, such as atherosclerosis, cancer, diabetes, neurodegenerative disease, and virus infection. It is hoped to provide a reference for disease diagnosis, monitoring, prognosis, and drug development.

## 1. Introduction

Glycosaminoglycan is a kind of unbranched linear anionic polysaccharide composed of 10 to 200 repeating disaccharide units [1]. Each disaccharide unit consists of one hexuronic acid (except keratan sulfate) and one hexosamine. The hydroxyls and/or acetyls on these disaccharide units are substituted by sulfate groups to varying degrees. Modification and isomerization of sugar residues leads to greater molecular diversity of GAGs. According to the structural characteristics of the disaccharide unit, GAGs can be divided into four classes: heparin/heparan sulfate (HP/HS), chondroitin/dermatan sulfate (CS/DS), keratan sulfate (KS), and hyaluronan (HA) [2].

The disaccharide structure of HP/HS isα-L-IdoA/β-D-GlcA (1→4) α-D-GlcNS/GlcNAc (1→4). HP/HS is typically 2-*O*-sulfated (2-*O*-S) at uronic acid residues and *N*-sulfated (*N*-S), 6-*O*-sulfated (6-*O*-S), and 3-*O*-sulfated (3-*O*-S) at glucosamine (GlcN) residues. Compared with HS, HP has a higher iduronic acid (IdoA) content (>70% in uronic acid) and sulfate/disaccharide ratio (about 2.3 sulfate groups per disaccharide in HP compared to 0.8 sulfate groups per disaccharide in HS) [3]. HP mainly exists in the intracellular granules of mast cells, and it possesses significant anticoagulant and other activities, such as antiviral and antitumor metastasis [4]. HS is in the extracellular matrix (ECM) and plays an important role in cell growth, immune response, tissue homeostasis, and embryonic development [5,6]. The disaccharide structure of CS/DS isβ-D-GlcA/α-L-IdoA (1→3) β-D-GalNAc (1→4). The 4-*O,* 6-*O* positions on GalNAc and 2-*O* position on Glc/IdoA can be sulfated. In addition to playing a structural role in organs and tissues (e.g., skin and cartilage), CS/DS was, in recent years, found to be involved in important biological processes [7,8] such as tumorigenesis and metastasis [9], nervous system development [10,11], and immune regulation [12]. KS differs from HS and CS/DS in that it takesβ-D-Gal (1→4) β-D-GlcNAc (1→3) as a disaccharide unit with 6-*O*-S at galactose and glucosamine [13]. KS is widely distributed in the body, such as in the eyes, brain, cartilage, and epithelial tissue. Therefore, KS has multiple physiological functions and plays an important role in the regulation of neuronal charge and ion gradients [14], cell adhesion, proliferation, and differentiation [15]. HA is the only nonsulfated polysaccharide in the GAG family, with a repeating disaccharide unit ofβ-D-GlcA (1→3) β-D-GlcNAc (1→4). HA is the largest-molecular-weight GAG (from 10^5^ Da to 10^7^ Da) and is abundant in connective tissues, such as synovial and vitreous fluid. HA not only provides compressive strength, lubricity, and hydration in the ECM, but also modulates cell adhesion metastasis, inflammation, and tissue homeostasis [16].

Most GAGs usually do not exist alone in vivo. In addition to HA, GAGs are linked to core proteins to form *N*-linked/*O*-linked proteoglycans (PGs) [17]. For example, HS and CS/DS chains are linked to the serine of core proteins via a tetrasaccharide unit of GlcA (1→3) β-Gal (1→3) β-Gal (1→4) β-Xyl (1→4). In fact, they are products of posttranslational modifications (glycosylation) of proteins. PGs are widely present on the cell surface, extracellular matrix (ECM), and basement membrane (BMs) (Figure 1). They participate in various life activities. GAGs/PGs are considered to be the most complex and informative biomolecules in organisms due to the differences in glycosidic bond types, polymerization, sulfation patterns, monosaccharide types, and core proteins [18]. Therefore, GAGs/PGs are indispensable for exerting the normal physiological functions of cells. They not only act as traditional supporting structures, but also interact with many extracellular signaling molecules, binding proteins, and enzymes to participate in different biological processes [17,19,20,21].

It is well known that the structure and distribution of GAG/PGs are different in various cells/tissues because they are affected by tissue-specific expression, activity level, and specificity of biosynthetic and degradative enzymes [22,23,24]. The decoration patterns of GAG chains influence and determine their specific interactions with natural protein ligands. This also reflects the physiological or pathological state of cells or tissue to some extent. In this review, we focus on the structural alterations and functions of GAGs/PGs in major human diseases (atherosclerosis, cancer, diabetes, neurodegenerative disease, and virus infections), hoping to provide a reference for disease diagnosis, monitoring, prognosis, and drug development.

## 2. GAGs in Atherosclerosis

Cardiovascular disease (CVD) is the leading cause of death in many countries [25]. CVD can be caused by many factors, such as genetics, diseases (e.g., diabetes [26,27]), and an unhealthy lifestyle (e.g., alcoholism and smoking). In addition, oxidative stress [28] and inflammation [29,30] are thought to be involved in the pathogenesis of CVD. Atherosclerosis is the most prevalent and clinically significant CVD, and it is closely related to other diseases, such as retinopathy, neuropathy, and nephropathy [31,32,33]. Human atherosclerosis develops in different stages, usually including intimal hyperplasia and lipid accumulation, foam cell formation, plaque formation and growth, plaque rupture, and thrombosis [34]. GAGs/PGs are the main components of the ECM in vascular wall cells, such as endothelial cells, smooth muscle cells (SMCs), and adventitial fibroblasts. Therefore, GAGs/PGs play a crucial role in regulating vascular permeability and maintaining homeostasis of the vascular environment [35,36].

CS and DS are the major GAG components of arterial vessels. The most abundant CSPG in the vascular ECM is versican [37,38]. DSPGs mainly include decorin and biglycan, which belong to the small leucine-rich repeat family. GAG chains of CS/DSPGs perform important functions in binding to lipoproteins and regulating elastin synthesis [39]. Moreover, CS/DSPGs have been found to be significantly elevated in early atherosclerotic lesions. They can promote lipid retention and accumulation, which is considered to be the initial factor in the development of atherosclerosis [40,41]. For instance, versican and biglycan strongly bind both native LDL and OxLDL [42]. Decorin binding collagen type I can promote LDL binding and enhance lipoprotein accumulation in the vascular wall [43]. Moreover, LDL–GAG complexes can be internalized by macrophages and SMCs [12,44] together with proliferation and migration of SMCs. This leads to intracellular lipid accumulation, foam cell formation, and intimal hyperplasia, which are considered hallmarks of early atherosclerosis [45]. In addition, CS chains can interact with elastin-binding protein (EBP) on the cell surface, inhibit the assembly of elastic fibers, and increase the rigidity of the vascular wall [46]. The CS/DS chain length, charge density, and sulfation pattern influences CS/DSPG binding to lipoproteins [47]. A study showed that the domains of 6-*O*-S or 4-*O*-S galactosamine (GalN) underlie CS/DS binding to lipoproteins [48]. The content of CS containing 4-*O*-S or 6-*O*-S in the aorta is 30% higher than in other arteries, which binds lipoproteins more tightly [49]. Theocharis et al. demonstrated an increase in both total CS content and the ratio of 6-*O*-S/4-*O*-S disaccharides in type II atherosclerosis arteries. This also denotes an increase in the CS/DS ratio, since CS and DS are the main providers of 6-*O*-S disaccharides and 4-*O*-S disaccharides, respectively [50]. In addition, transforming growth factor (TGF-β) in atherosclerotic vessels prolongs CS chains in arterial smooth muscle cells (ASMCs), thereby increasing versican binding to LDL [51]. Therefore, CS/DS can promote the development of atherosclerosis under pathological conditions.

The major HSPG in vascular endothelial cells and SMCs is perlecan [52]. Earlier studies showed that vascular HSPG levels are decreased and negatively correlated with cholesterol levels in atherosclerosis patients [53,54,55]. Thus, HSPG shows the opposite trend to CS/DSPGs in atherosclerosis. The reduction in HSPG could increase the binding of lipoprotein A to the ECM in endothelial cells, which may be related to the fact that OxLDL encourages endothelial cells to produce more heparinase [56]. Heparanase leads to abnormal HS degradation, which, in turn, results in increasing vascular endothelial permeability and SMC migration. This suggests that rising HSPG synthesis may protect against atherosclerosis [57]. Moreover, as mentioned earlier, large numbers of macrophages are recruited in atherosclerotic lesions. HSPG in macrophage ECM has also been implicated in atherogenesis. Asplund et al. found that HS depletion on the surface of human macrophages (HMDM) under hypoxia enhanced cell motility and accelerated plaque formation [58].

Deposition of lipids in the vascular wall promotes proliferation and migration of SMCs, where HA appears to play a major role [59]. HA in the arterial wall is mainly produced by SMCs, and synthesis increases in atherosclerosis [60]. A study showed that HA accumulation could promote SMC metastasis via ERK1/2 modulation of the CD44 signaling pathway, resulting in intimal hyperplasia [61]. HA also binds to LDL and is more readily internalized by macrophages than native LDL [62]. Low-molecular-weight hyaluronic acid (LMW-HA) is considered to induce inflammation in vivo, whereas high-molecular-weight HA (HMW-HA) does not [63,64]. Tabata et al. found that LMW-HA, which is abundantly produced in atherosclerotic lesions, participates in the inflammatory mechanism of atherosclerotic plaque formation by promoting monocyte migration and foam macrophage differentiation through binding to CD44 [65]. In addition, HA can interact with versican to remodel the ECM of diseased SMCs and promote the proliferation and migration of SMCs [44]. In short, LDL internalization impels macrophages to release more cytokines and growth factors, which changes the behavior of SMCs. Meanwhile, overexpression of HA and its receptor CD44 further aggravated cell migration and intimal hyperplasia.

In the process of atherosclerosis, the content of total CS and 6-*O*-S of CS are increased, enhancing binding to lipids, and LDL-GAG complexes can be internalized by macrophages and SMCs, promoting the formation of sclerotic plaques. In addition, HS depletion in the ECM at the lesion resulted in increased vascular endothelial permeability, promoting inflammatory cell infiltration and cell migration. The synthesis of LMW-HA is increased, which is involved in the inflammatory mechanism of atherosclerotic plaque formation. In conclusion, GAGs/PGs participate in ECM remodeling and play multiple roles in regulating immune adhesion, as well as promoting lipid accumulation, intimal hyperplasia, and thrombosis. Therefore, vascular GAGs/PGs have potential as a target for the prevention of atherosclerosis.

## 3. GAGs in Cancers

Cancer is the second leading cause of death worldwide [66,67]. Cancer possesses biological characteristics, such as uncontrolled cell differentiation and proliferation, invasion, and metastasis. Its occurrence is a multifactorial and multistep complex process [68]. Cancer is defined as many types, such as liver, lung, breast, and ovarian cancer, depending on where the cancer is located. During cancer development, growth, metastasis, and invasion require specific interactions between tumor cells and the tumor microenvironment (TME) [69,70,71]. GAGs/PGs, as the critical effectors of the cell surface and TME, are involved in tumor growth and metastasis through interacting with growth factors, growth factor receptors, and cytokines [69,72,73,74]. Importantly, GAGs/PGs play a vital role in cancer regulation in terms of types, molecular weight, distribution, and fine modification. Thus, GAGs/PGs can become potential targets for anticancer therapy [75,76].

HSPG (mainly perlecans, syndecans, and glypicans) are considered central molecules regulating cell behavior and cancer progression [77]. HSPGs are differentially expressed in diverse cancers. For example, in breast cancer, perlecan was absent in epithelial cell BM, while being markedly upregulated in stroma. Furthermore, plasma perlecan level was significantly higher in estrogen receptor (ER)^+^ patients than ER^−^ patients [78]. Perlecan expression is increased in invasive and metastatic prostate cancer cells [79]. Syndecan-2 was significantly increased in well-differentiated neuroendocrine tumors (NETs) and significantly decreased in poorly differentiated NETs. Glypican-5 was overexpressed in high-grade tumors with epithelial differentiation, but not in tumors with neuroendocrine phenotype [80]. Indeed, the sulfation pattern of GAG chains is also strongly related to cancer type and differentiated degree [81,82,83,84,85,86]. For instance, Weyers et al. examined changes in the GAGs of fatal and nonfatal breast cancer tissues. The GAG length increased by approximately 15% in tumor tissue compared to normal tissue. Both the 6-*O*-S CS and the total sulfation of HS increased. Compared to nonfatal breast cancers, the sulfation degree of HS, particularly 6-*O*-S, was decreased in fatal breast cancers, whereas the proportion of non-sulfated disaccharides was increased [87]. In addition, the expression profile of HS in cancer and its role in cancer regulation have been detailed in a recent review [88].

Many studies have indicated that *N*-S, 2-*O*-S, and 6-*O*-S of HS play an important regulatory role in tumor metastasis and invasion, especially 6-*O*-S [89]. For example, fibroblast growth factor/fibroblast growth factor receptor (FGF/FGFR) promotes cardiovascular generation and endothelial cell repair, playing a key role in the regulation of lesion metabolism [90] (Figure 2). Therefore, FGF/FGFR is widely regarded as a potential target for antitumor therapy [91,92]. HS promotes cell signal transduction by binding to FGF2/FGFRs to form ternary complexes [93,94]. In this process, HS binding to FGF2 requires the *N*-S of GlcN and 2-*O*-S of IdoA [95]. In addition to *N*-S and 2-*O*-S, HS requires 6-*O*-S of GlcN to bind to FGFR. Similarly, IdoA with 2-*O*-S and GlcN with 6-*O*-S and *N*-S are also essential for binding HS to FGF1 [96]. Specific sequences for HS binding to FGF family proteins were further identified by Kreuger et al. They found that HS octasaccharides binding to FGF1 contained an IdoA (2S)–GlcNS (6S)–IdoA (2S) trisaccharide sequence, and HS binding to FGF2 contained an IdoA (2S)–GlcNS–IdoA (2S) trisaccharide sequence [97]. Schultz et al. determined the fine structure of HS binding to FGF1_2_–FGFR1c_2_ from chemically enzymatically synthesized HS octasaccharide, and they pointed out that the *N*-S of the nonreducing terminal residue is essential for binding [98]. In addition, HS on the tumor cells surface showed a stronger affinity for NT4 (a tetrapeptide) compared with other GAGs [99,100]. NT4 can target cell lines of different human cancers; therefore, it may serve as a carrier for the delivery of anticancer drugs or tumor imaging tracers [101,102,103,104,105]. More importantly, NT4 may inhibit the migration of pancreatic cancer cells, as well as the growth factor-induced invasion of breast cancer cells, by binding to HS [99,106]. Brunetti et al. demonstrated the decisive role of sulfate groups on HS for binding to NT4 and determined that possible binding sequences include repeated disaccharide units of uronic acid with 2-*O*-S and GlcN with 6-*O*-S and *N*-S [107]. Interestingly, Liu et al. treated tumor cells with heparinases I and III, respectively. They demonstrated that HS on the cell surface has specific sequences for both tumor activation and inhibitory activity [108]. It was proven that the tumor metabolic regulatory effect of HS is closely linked to its specific sulfation patterns.

CS/DS is equally important during tumor cell proliferation, migration, adhesion, and invasion [109]. In some cancer tissues, the CS content is increased [110,111,112,113,114]. For example, glioblastoma multiforme (GBM) tumors with increased CSPG content accounted for 65% of the total number of samples through clinical studies by Tsidulko et al. [115]. CSPG, such as versican, is highly expressed in breast cancer, which is associated with an unfavorable prognosis [116]. For a few cases, such as invasive gliomas, several studies have shown that the GAG chain of CS/DSPGs in the tumor ECM is severely absent [117,118,119,120,121]. Abnormal glycosylation may be associated with isomeric conversion and lyase activity of CSPGs [120,121,122,123]. The change of DS depends on cancer type. DS levels are elevated in liver cancer [124], lung cancer [125], gastric cancer [126], pancreatic cancer [127], and colorectal cancer [113]. Although the content and distribution of CS/DS are heterogeneous in tumor tissues, several studies have shown that 6-*O*-S and non-sulfated disaccharide levels are increased in some tumors, while 4-*O*-S disaccharide levels are decreased [128,129]. For example, increased CS content is observed in pancreatic cancer, characterized by a significantly enhanced expression of 6-*O*-S and non-sulfated disaccharide units [127,130]. Prostate cancer has elevated 4-*O*-S CS content in the ECM, which may be due to inhibited androgen receptor (AR) signaling, thus resulting in increased 4-*O*-sulfotransferase CHST11 expression [131]. CS testing of human gastric cancer tissues revealed a 10-fold increase in 6-*O*-S and non-disaccharide units, while 4S disaccharides were correspondingly decreased [132]. In addition, CS with 4, 6-*O*-S (CS-E) is increased in a variety of cancers. A high expression of CS-E in the ECM of ovarian adenocarcinoma enhances vascular endothelial growth factor (VEGF) mediation [133]. The proportion of Δ4.5HexA-GalNAc-4, 6-*O*-disulfate was higher in highly metastatic lung cancer cell lines than in low metastatic cell lines [134]. The results suggest complex changes in the sulfuryl modification of CS/DS during carcinogenesis. A possible reason is that, on the one hand, aberrant expression of the PG core protein leads to changes in the type and structure of the linked GAG chain [135]. On the other hand, CS/DS synthesis or modification is mediated by abnormalities in enzyme activity or levels [136,137,138]. However, the reduction in DS in tumor tissue may be more conducive to cancer development and metastasis. Therefore, it has been suggested that DS has antitumor activity. For instance, DS/DSPG has an inhibitory effect on the proliferation and migration of certain osteosarcoma and melanoma cell lines [139,140,141]. However, there seems to be controversy regarding the role of DS in cancer. DS was found to promote proliferation of esophageal squamous cells [142]. Therefore, the role of DS in cancer needs further investigation.

HA has the dual properties of tumor promotion and tumor inhibition. LMW-HA predominates in normal tissues and is essential for maintaining tissue homeostasis. Studies have shown that HA content increases in many types of human cancers [143,144,145]. Moreover, HA molecular weight decreases in the TME [146,147,148]. LMW-HA stimulates the expression of chemokines and growth factors, as well as promotes tumor cell adhesion and migration [149,150]. The role of LMW-HA in cancer is associated with HA receptors (e.g., CD44, and HA-mediated mobility receptor (RHAMM)) [151,152]. In addition, HA has been reported to be degraded by hyaluronidase into smaller oligosaccharide fragments, inducing the division of CD44, thereby enhancing tumor cell viability in breast, ovarian, and glial tumors, and colon cancers [153,154].

For other GAGs, Yukinari et al. found that highly sulfated KS was overexpressed in malignant astrocytic tumors using histochemistry [155]. 

In short, GAGs/PGs are critical regulators for cancer cell proliferation and metastasis. Although changes in GAGs/PGs vary widely in different types and stages of cancer, overall HS expression is upregulated in cancer cells relative to normal cells. As important signaling molecules on the cell surface, HS overexpression increases the communication between cancer cells and the external environment, which supports the characteristics of cancer cells that are prone to proliferation and metastasis. During this period, *N*-S, 2-*O*-S, 6-*O*-S of HS play an important regulatory role. CS/CSPGs are also upregulated in some tumors, and at the same time, they are accompanied by increased 6-*O*-S levels and decreased 4-*O*-S levels. DS/DSPG is detected in some tumors, which suggests that DS may have antitumor activity. LMW-HA increases in many types of cancer and promotes tumor cell adhesion and migration. Thus, GAGs/PGs play a crucial role in tumor cell activity. In addition, GAGs perform their functions by interacting with proteins in the body. Therefore, the development of some GAGs analogues to inhibit their interactions may become another approach for cancer treatment [156].

## 4. GAGs in Diabetes Mellitus

Diabetes mellitus (diabetes, DM) is a metabolic disorder caused by decreased insulin levels due to autoimmune β-cell destruction (T1D) and/or insulin resistance (T2D) [127,157]. Diabetes is one of the fastest-growing diseases, with approximately 415 million patients worldwide in 2015, and this number is expected to grow to 693 million by 2045 [157,158]. The development of diabetes can cause systemic diseases, such as cardiovascular disease, diabetic nephropathy, retinopathy, and neuropathy. The ECM of various tissues or organs acts as the first line of defense against pathological factors, and ECM remodeling plays a crucial part in the development of diabetes and complications [159].

In normal pancreatic β-cells, high levels of HSPG are necessary for cell survival. In normal metabolism, HSPG can not only prevent the invasion of immune cells and prevent the degradation by heparanase, but also act as a nonenzymatic antioxidant that can scavenge ROS in a timely manner and avoid oxidative damage to cells [160,161]. In studies of T1D, HSPGs such as Col18 and syndecan-1 were found to be highly expressed in normal human islets. Importantly, highly sulfated HS was specifically expressed in normal human β-cells, while highly sulfated HS was expressed in α-cells [162]. Further studies revealed that HS in β-cells largely contain 2-*O*-S, 6-*O*-S, and *N*-S, whereas in α cells, HS contains *N*-acetyl, *N*-S, and 2-*O*-S, with fewer 6-*O*-S. Moreover, HS mediates cell activity through FGF/FGFRs [163] (the effect of HS on the FGF/FGFR signaling pathway was presented in the Section 3). When β-cell dysfunction occurs, it is often accompanied by inflammation such as pancreatitis; leukocytes infiltrate islet cells, release heparanase to degrade HSPG, gradually reduce islet HS, and injure β-cells [160,164,165]. Low-sulfated HS decreases β-cell proliferation and viability by mediating FGF/FGFR signaling [163]. At the same time, the decrease in the content of highly sulfated HS renders β-cells highly susceptible to oxidant-mediated damage, while high levels of endogenous ROS can depolymerize HS [166,167]. In addition, for HSPG alterations in T2D, Simeonovic et al. proposed the “ER stress” model: β-cell endoplasmic reticulum (ER) homeostasis is disrupted during the pathological process of T2D, resulting in ER luminal misfolded protein accumulation and ER stress [168,169]. ER stress initiates the unfolded protein response (UPR), which alleviates stress by increasing chaperones for protein folding and decreasing normal protein synthesis [170]. This impaired HSPG core protein synthesis further hinders HS synthesis. HS acts as a nonenzymatic antioxidant in β-cells, and its lower expression increases intracellular ROS and promotes oxidative stress, ultimately leading to the apoptosis of β-cells [171,172]. Thus, a high expression of highly sulfated HS is a hallmark of healthy β-cells. Alternatively, β-cells are metabolically disturbed under pathological conditions, and islet amyloid polypeptide (IAPP) undergoes misfolded aggregation, resulting in deposition. IAPP is a hormonal peptide implicated in T2D pathogenesis and progression [173]. GAGs were found to be involved in the deposition of IAPP within or around pancreatic β-cells [174]. Castillo et al. investigated the interaction of different sulfated GAGs, including perlecan, with amyloid in vitro and found that GAG sulfation patterns affected amyloid fibril formation. The order of impact was heparin > *N*-desulfated acetylated heparin > fully desulfated *N*-sulfated heparin > fully desulfated *N*-acetylated heparin [175]. Recent studies have shown that PGs on the surface of islet cells, especially HS/HSPGs, are able to promote islet amyloid deposition and IAPP-induced cytotoxicity, thereby accelerating T1D progression [176,177]. Nevertheless, there is little information on other GAGs involved in islet cell lesions. Only a few reports mentioned that HA levels were significantly elevated in immune cells both inside and outside islet cells and at inflammatory sites in T1D [178,179].

Diabetic nephropathy (DN) is one of the most important complications of diabetes and is characterized by proteinuria [180]. Earlier studies have shown that reduced HS levels in diabetic glomerular basement membranes (GBM) are associated with proteinuria [181]. The altered HS sulfation pattern of GBM in diabetic nephropathy may be due to increased heparanase levels [182] or altered sulfatase regulation, and it has been demonstrated that 6-*O*-S of HS plays an important role in maintaining the glomerular filtration barrier [183]. In another study, HS was found to have less *N*-S in the GBM of diabetic rats compared to the normal group [184]. Paradoxically, however, some researchers have demonstrated that the structure of glomerular HS is not affected in diabetic rat models [185,186]. Compared with HS, CS/DS alterations were more significant in the kidney. In diabetic rats, renal CS/DS content decreased, accompanied by a decrease in the degree of sulfation, particularly 4, 6-*O*-sulfated GalN content [187]. Another study described CS alterations in the kidney in more detail. Reine et al. found that 4-*O*-disaccharide sulfate significantly decreased from 65% to 40%, whereas 6-*O*-S disaccharide decreased from 11% to 6% and non-sulfated disaccharide increased from 21.5% to 51% in the renal cortex of diabetic db/db mice [186]. Alterations in CS/DS structure in the kidney of patients with DN have an impact on the composition and function of the ECM in which it is located. Glomerular filtration of urine is also compromised. A study from diabetic patients with T2D found that the contents of total GAGs, CS/DS, and HS in urine were significantly higher than those in healthy subjects [188]. Through quantitative analysis, it was found that 6-*O*-S and the 6-*O*-S/4-*O*-S ratio in the urine of diabetic patients with microalbuminuria were significantly increased compared with the healthy group [189]. In addition, CS/DS in erythrocytes was investigated in a diabetic rat model by Srikanth et al. They found a twofold increase in CS/DS content, mainly consisting of 4-*O*-S disaccharide units and a small number of non-sulfated disaccharides [190]. This suggests that the level of GAG in urine and blood or the ratio between them has the potential to be a diagnostic indicator of diabetes. However, the alteration of GAGs was mainly detected in advanced diabetic nephropathy, the elevation of GAG levels in urine/serum in the early stage of the disease is not very clear.

Diabetes begins with islet cell dysfunction, but persistent hyperglycemia causes a variety of complications. The ECM remodeling exhibited by various complications is also different. Therefore, finding a link between these GAG alterations is expected to provide useful information for the treatment of diabetes and its complications.

## 5. GAGs in Neurodegenerative Disease

Neurodegenerative disease is a type of disease caused by the degenerative loss of neurons in the brain and spinal cord. Its occurrence may be multifactorial (including genetics, oxidative stress, neuroinflammation, mitochondrial damage, and abnormal protein folding) [191], but its pathogenesis has not yet been determined [192,193]. Neurodegenerative diseases are classified according to pathological features, mainly including Alzheimer’s disease (AD) and Parkinson’s disease (PD), Huntington’s chorea, and amyotrophic lateral sclerosis. Protein misfolding and aggregation is one of the features of neurodegenerative diseases, such as β-amyloid (Aβ) and phosphorylated tau in AD, and α-synuclein (α-syn) aggregation and fibrogenesis in PD [194,195]. Neuronal cells are the main site of lesions in this kind of disease. Perineuronal nets (PNNs) are specialized extracellular matrices of neurons. They are mainly composed of CSPGs and HA, and they play an important part in the normal function of the central nervous system [196,197,198]. At the same time, changes in PNN composition also reflect a variety of neuropathological states [199].

The most abundant PG found in the nervous system is CSPG [200], and its sulfation pattern is thought to be directly linked to pathological development [196,201]. PGs in most neuronal PNNs mainly contain versicans, aggreicans, neuroicans, and brevicans [199]. It has been shown that CS-C (6-*O*-S) has the highest proportion in the early stages of brain development and is gradually replaced by CS-A (4-*O*-S) as the brain develops and matures [202]. This change illustrates the important role of CS-A in stabilizing PNNs and limiting neuronal plasticity [203,204]. However, this state shifts in AD. A recent study showed that CS quantification in prefrontal neocortical (middle frontal gyrus) samples from AD patients showed an increased CS-C and CS-E content and decreased non-sulfated CS disaccharides, but total GAG levels did not change [201]. In addition, a few reports have shown that CSPGs are also involved in the formation of amyloid precipitation [205,206]. Early studies certified that 4-*O*-S and 6-*O*-S CS are found in neurofibrillary tangles (NFTs) of AD patients, while only 4-*O*-S CS is found in senile plaques (SPs) [207]. In addition, in multiple sclerosis (MS), inflammatory cell infiltration, demyelination, and axonal injury result in sclerosing lesions in the white matter [208]. Studies have shown that CS/DSPGs are the main PG components of plaques [209]. It was found that CSPGs (versican, neurocan, and aggrecan) and DSPGs were mainly located at the edge of active plaques, while the content of CSPGs in the active center of MS plaques was significantly decreased, possibly due to the internalization of PGs in PNNs by their foam macrophages together with myelin [208]. Moreover, increased CSPG content at the plaque edge can inhibit oligodendrocyte precursor cells (OPCs) [210,211,212,213], which are essential for reconstructing myelin sheaths and protecting neurons. Further studies showed that the CS chain of CSPG exerts its inhibitory effect on OPCs mainly through 4-*O*-S and 6-*O*-S disaccharide units [214].

Although HS is less abundant in central neurons, it is the most studied PG in neurodegenerative diseases. HS has been demonstrated to be involved in abnormal protein accumulation within and around neurons, such as amyloid plaques and neurofibrillary tangles [197,206,215]. Extracellular Aβ plaques and intracellular NFTs are neurotoxic in AD, ultimately leading to neuronal loss [216]. In the brains of AD patients, PGs were more abundant in areas with amyloid plaques and neurofibrillary tangles. For example, relative to healthy individuals, the total PGs increased 1.6-fold in the AD hippocampus and 3.4-fold in the superior frontal gyrus (superior gyrus frontalis). Among them, HSPGs increased the most [217]. As described earlier, GAGs promote amyloid fibril formation, as determined by the chain type and sulfation pattern [175]. Staining of occipital neocortical and hippocampal tissue from AD patients revealed that fibrillar Aβ plaques and nonfibrillar Aβ plaques contained high levels of *N*-sulfated HS, while *N*-sulfation was very low in nonfibrillar Aβ plaques [218]. Lindahl et al. identified critical sites for the binding of heparin sulfate to Aβ fibrils containing 2-*O*-S IdoA and *N*-S from the human cerebral cortex, whereas the binding of Aβ monomers requires 6-*O*-S on GlcN residues [219]. The sequence of heparin oligosaccharides interacting with Aβ was determined using 2D NMR and molecular simulation docking techniques by Zhou et al. [220]. They found that the binding motif of HS is HexA–GlcNS–IdoA2S–GlcNS6S; IdoA and 6-*O*-S are required for binding [220]. This is consistent with previous findings. At the same time, they also identified the amino acid sequence of the Aβ-binding site as V12HHQKL17 using hydrogen–deuterium exchange mass spectrometry (HDX-MS) [220]. Coincidentally, the HS sequence profiles identified in the above studies are identical to FGF-2-binding sites [219]. Thus, the binding of HS to Aβ fibrils may competitively inhibit FGF2-mediated neuroprotection. In addition, GAGs can interact with tau protein, stabilize tau conformation, and promote its phosphorylation [221]. A recent study demonstrated that 3-*O*-S in HS enhances HS and tau binding and promotes tau transport across membranes [203]. Thus, tau is one of proteins that recognizes 3-*O*-S of HS. α-Synuclein (α-syn) is a major pathogenic protein in PD and a major component of LBs. HS interacts with α-syn and influences α-syn fibril conformation choice. Liu et al. found that HSPG (agrin) accelerated the formation of α-syn fibrils and induced α-syn protein β-sheets in an HS-dependent manner, enhancing the insolubility of α-syn [222]. A recent study determined the atomic structure of α-syn fibrils formed by heparin involvement and discovered a novel folding mode of α-syn, the “Z” fold [223]. These results indicate that sulfated GAGs play a crucial role in protein aggregation. Furthermore, Ishe et al. confirmed that, in neuronal cells, internalization of α-syn aggregates strongly depends on the cell surface HS and is associated with their total sulfation level [224]. These results indicate that HS acts on α-syn similarly to Aβ. In addition to HS playing an important role in PD progression, the presence of CS (4-*O*-S and 6-*O*-S) with different degrees of sulfation in LBs was reported by DeWitt et al. [225].

At present, the pathogenesis of most neurodegenerative diseases is still unclear. However, it is certain that GAGs are indeed involved in the development and progression of the disease. Therefore, an in-depth study of GAGs is an important aspect of mechanistic studies of neurodegenerative diseases.

## 6. GAGs in Virus Infection

Viruses are the main pathogens threatening human health, and they are characterized by species diversity, strong infectivity, diverse transmission routes, and easy variation. The worldwide spread of COVID-19 disease caused by severe acute respiratory syndrome-associated coronavirus 2 (SARS-CoV-2) in recent years is a living example. Furthermore, Ebola virus, HIV virus, SARS virus, and hantavirus have a high mortality. A common feature of viruses infecting humans is that they enter host cells by binding to cell-surface receptors. GAGs in the ECM have been found to play an important role in regulating immune defense and pathogenic mechanisms [226,227,228]. For instance, when SARS-CoV-2 infects host cells, HS acts as a cofactor involved in the interaction between the SARS-CoV-2 spike glycoprotein receptor-binding domain (RBD) and angiotensin-converting enzyme 2 (ACE2) [229,230,231]. In addition, dengue virus (DENV) [232,233], hepatitis C virus (HCV) [234], herpes simplex virus [235,236], human papillomavirus (HPV) [237], arboviruses [238], respiratory syncytial virus (RSV) [239], and monkeypox virus (MPXV) [240] have all been found to invade host cells by binding HS and/or CS/DS in the ECM through receptor proteins. Specific sulfation patterns of GAGs are critical for viral adsorption and invasion. For example, DENV-secreted NS1 protein accumulates on infected cell membranes and interacts with HS and CS-E on the cell surface, ultimately leading to selective vascular leak syndrome [232]. Kim et al. found that *N*-S, 2-*O*-S, and 6-*O*-S in HS and 6-*O*-S, 2-*O*-S, 3-*O*-S, and *N*-S in HP were critical for competitive binding to SARS-CoV-2 spike protein [241]. The amino-acid sequences of spike protein trimer-binding sites to GAGs are YRLFRKS, PRRARS, and SKPSKRS. Thus, HS is a potential competitive inhibitor against SARS-CoV-2 infection [241]. In parallel, Tiwari et al. confirmed that the presence of 3-*O*-S in HS contributes to the recognition and binding of SARS-CoV-2 spikes in vitro [242]. This idea was confirmed in another report, where 3-*O*-sulfotransferase 3B overexpression and glycocalyx sulfation degree were too low to promote SARS-CoV-2 infection under pathological conditions [243]. However, there are few reports on the alterations of GAGs in virus infection in vivo [244]. In brief, GAGs in cellular ECM execute critical roles in regulating viral adhesion and invasion. 

## 7. GAGs in Other Diseases

GAGs/PGs, as the main component of ECM, are involved in the development and progression of almost all human diseases. Pulmonary diseases have a high morbidity and mortality, such as pulmonary fibrosis, asthma, and chronic obstructive pulmonary disease (COPD) [245,246]. In idiopathic pulmonary fibrosis, CS/DS, HA, and the CS/DS ratio increased significantly. CS/DS increased in 4-*O*-S, 6*-O*-S, and 2*-O*-S disaccharide units and decreased in non-sulfated disaccharides, resulting in significant increases in sulfated levels. Similarly, significant increases were observed in *N*-S, 2*-O*-S, and 6*-O*-S disaccharides of HS, particularly the UA2S–GlcNS6S unit [247]. In addition, significant increases in PG (versican, biglycan, and decorin) content at the lesion were observed in biopsy specimens from asthma cases [248]. In COPD, although HS increased significantly, its sulfation pattern was related to the COPD stage, while CS/DS did not change significantly. The 2-*O*-S and *N*-S of HS increased during the fourth phase of COPD, while 6-*O*-S did not change [249]. GAG expression is elevated in cystic fibrosis [250]. Moreover, abnormal sulfation of GAG can be found in bronchial epithelial cells of patients [251]. Kim et al. investigated CS expression in lupus erythematosus (LE) and dermatomyositis (DM) [252]. Expression of 4-*O*-S CS was found to be increased only in discoid lupus erythematosus (DLE) and DM, whereas 6-*O*-S CS was found to be significantly increased in dermal endothelium with DM [252].

Kidney stones are a common urinary disorder. Due to metabolic abnormalities, crystals such as uric acid and calcium oxalate (CaOx) accumulate in the renal pelvis or calyces of patients’ urine, resulting in stone formation. It has been found that the type and content of GAGs in urine may be closely related to the formation process of renal calculi [253,254,255]. Some earlier studies demonstrated that CS and HS inhibited the formation of kidney stones, while HA promoted stone formation and growth [256,257,258]. Dissayabutra et al. tested GAGs in urine from familial urolithiasis cases and found that total sulfated GAGs, CS, and HS contents in urine were all decreased, while HA content was increased, and the proportion of HS in total sulfated GAGs was increased [259]. Jappie et al. detected urine from healthy white and black South Africans. Blacks were found to have higher CS levels in urine than whites (kidney stones were significantly more prevalent in whites than blacks in South Africans), suggesting that higher CS levels may inhibit kidney stone formation [260]. These findings are consistent with previous conclusions. However, a recent study found that HA inhibited aggregation of CaOx crystals in artificial urine, but did not have any effect on the crystalline properties of CaOx in real urine. In addition, it has been pointed out that the regulation of nucleation and growth of crystals in urine is the result of the combined action of various GAGs [261].

In human diseases, inflammation is deemed to be the “source of all diseases”. The human body is an organic system, and there are tied connections between various diseases. A normal inflammatory response is beneficial for the human body, but chronic inflammation is a common pathological basis for various diseases and may be an important factor in the increased morbidity and mortality of most diseases. For example, many cancers develop in the presence of chronic inflammation; cancer metastasis is also akin to an inflammatory response, and even many inflammatory cells participate in and assist in the metastasis of cancer cells. Changes in the inflammatory response, from short to long term, can lead to the collapse of immune tolerance and lead to major alterations in the physiology of all tissues and organs, as well as normal cells. This increases the risk of noncommunicable diseases in humans. The ECM remodeling involved in inflammation, particularly the alterations and roles of HS, CS, and HA, were extensively discussed in several recent articles [262,263,264,265].

## 8. Prospects and Challenges

GAGs/PGs are the most complex biological macromolecules in the human body, and their role in cellular life processes is justifiable. Various GAGs function in vivo by interacting with proteins via ionic, hydrogen, and hydrophobic bonds [3,266]. The structural basis for these interactions is a specific sulfation sequence in the GAGs’ sugar chain. Because GAG biosynthesis is a non-template-driven process, their structures are stochastic and variable. GAGs also display structural alterations during pathology (the alterations of GAGs in different diseases are summarized in Table 1). Thus, complex sugar chains also have huge information density and extensive structural and functional heterogeneity. Over the last few decades, numerous studies were conducted on the structure and function of GAG sugar chains, as well as related regulatory enzymes, and their changes in time and space were identified, thus providing substantial theoretical support for the elucidation of disease mechanisms and the exploration of therapeutic targets.

However, there are still great challenges for the study of GAGs in human diseases. On the one hand, it has been debated whether the binding of GAG sequences to proteins is specific. However, there is increasing evidence that proteins are highly selective for GAG sequences. In addition to the examples listed here, classical binding of heparin pentasaccharides to antithrombin III has been demonstrated. However, most studies on GAG chains binding to proteins were performed in vitro; hence, whether they can truly reflect in vivo behavior needs further investigation. On the other hand, due to the high heterogeneity of the GAG molecular structure, the current structural analysis of GAG chains lags behind that of other biomacromolecules (proteins and DNA). Despite the rapid development of modern analytical techniques, determination of the fine structure of GAGs remains a challenging task [2]. This has caused great hindrance to in-depth studies of the structure–function relationship of endogenous GAGs in physiological or pathological conditions.

At present, GAGs have become another “life code” to be deciphered after nucleic acids and proteins, and the elucidation of the relationship between GAG structure and function is of great significance for the prevention of human diseases and the implementation of precision medicine.

## Figures and Tables

**Figure 1 polymers-14-05014-f001:**
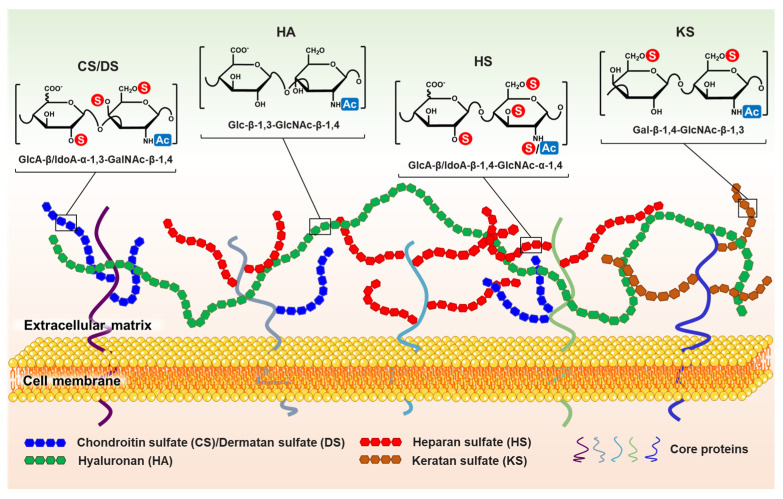
Distribution and structures (repeating disaccharide units) of GAGs in the extracellular matrix. CS/DS, HS, and KS are linked to core proteins, whereas HA is free. “
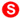
” and “
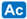
” indicate that the hydroxyl can be substituted by a sulfate group and acetyl, respectively. It is worth noting that the structural formula represents the maximum degree of sulfation for each GAG type. “
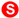
” only represents the possibility of being substituted by a sulfate group. Such as CS, they can be classified as CS-A (GlcA-GalNAc4S), CS-C (GlcA-GalNAc6S), CS-D (GlcA2S-GalNAc6S), CS-E (GlcA4S-GalNAc6S), etc. (Used with permission of Royal Society of Chemistry from ref. [1]; permission conveyed through Copyright Clearance Center, Inc).

**Figure 2 polymers-14-05014-f002:**
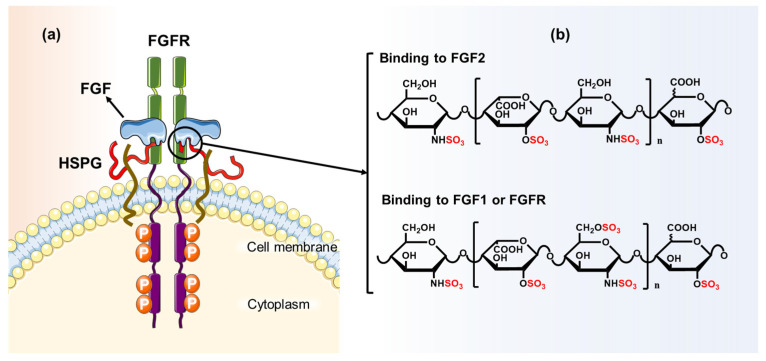
Schematic diagram of the ternary complex of FGF/HS/FGFR and the key sites for HS binding. (**a**) FGFR comprises extracellular Ig-like domains, intracellular tyrosine kinase domains, and transmembrane domain. Ig-like domains bind FGF with the assistance of HS to form ternary complex; (**b**) HS binding to FGF2 requires 2-*O*-S of IdoA and *N*-S of GlcN; HS binding to FGF1 and FGFR requires 2-*O*-S of IdoA and 6-*O*-S, *N*-S of GlcN; the *N*-S of the nonreducing terminal residue is also necessary for HS binding to FGF/FGFR. The increased expression of HS in most tumor cells enhances FGF/FGFR signal transduction, which is beneficial to tumor cell growth and angiogenesis. (The data of the HS sequence is cited from reference [96,97,98,99]).

**Table 1 polymers-14-05014-t001:** Alterations or roles of GAGs/PGs in different diseases.

Pathology Types	Study Objects (Cells/Tissues)	GAGs/PGs Alterations	References
Atherosclerotic type II	Aorta	Both the total CS content and the ratio of 6-*O*-S/4-*O*-S disaccharides in type II atherosclerosis arteries were increased	[50]
Atherosclerosis	Arterial smooth muscle cells	TGF-β prolongs CS chains in arterial smooth muscle cells and increases versican binding to LDL.	[51]
Atherosclerosis (symptomatic carotid stenosis)	Iliac arteries	The expression of perlecan gene decreased while versican gene remained unchanged.	[53]
Atherosclerosis and vascularrestenosis	Macrophages	Syndecan-1 protein level in macrophages was significantly decreased under hypoxia condition, and mRNA expression of key enzymes involved in HS biosynthesis in hypoxia cells was decreased. In addition, hypoxia also reduced the relative content of HS.	[58]
Atherosclerosis and vascularrestenosis	Aortic smooth muscle cells	HA and HA synthase are increased in senescent cells. HA accumulation promotes SMC metastasis via ERK1/2 modulation of the CD44 signaling pathway, resulting in intimal hyperplasia.	[61]
Atherosclerosis	Macrophages	LMW-HA induces macrophage/foam cell production and promotes atherosclerosis via the PKC pathway. LMW-HA also amplifies the migration of monocytes to inflammatory atherosclerotic plaques.	[65]
Breast cancer	Tumor tissue and plasma	Perlecan is absent in epithelial cell basement membrane while markedly upregulated in stroma. Furthermore, plasma perlecan level was significantly higher in estrogen receptor (ER)^+^ patients than ER- patients.	[78]
Prostate cancer		Perlecan expression is increased, which can the regulate sonic hedgehog signaling path.	[79]
Neuroendocrine tumors (NETs)	Tumor tissue	Syndecan-2 is significantly increased in well-differentiated NETs and significantly decreased in poorly differentiated NETs. Glypican-5 was overexpressed in high-grade tumors with epithelial differentiation, but not in tumors with neuroendocrine phenotype.	[80]
Breast cancer	Tumor tissue	The GAG length increased by approximately 15% in tumor tissue compared to normal tissue. Both the 6-*O*-S CS and the total sulfation of HS increased. Compared to nonfatal breast cancers, the sulfation degree of HS, particularly 6-*O*-S, was decreased in fatal breast cancers, whereas the proportion of non-sulfated disaccharides was increased.	[87]
Glioblastoma multiforme (GBM)	Tumor tissue	60–65% of GBM tumor samples showed increased levels of CS. A 1.5-fold increase in decorin, a 3-fold increase in biglycan and a 2-fold increase in serglycin. Only decorin levels were negatively associated with overall survival in GBM patients.	[115]
Pancreatic carcinoma	Tumor tissue	The total GAG level was increased by 4 times, HA increased 12 times, CS increased 22 times, DS increased 1.5 times. A significant increase in non-sulfate and 6-sulfate disaccharides of CS.	[127]
Pancreatic carcinoma	Tumor tissue	There are 27-fold and 7-fold increases in versican and decorin, respectively, compared with normal pancreases. The expression of 6-O-S and non-sulfated disaccharide units are enhanced.	[130]
Prostate cancer	Tumor tissue	Prostate cancer has an elevated 4-*O*-S CS content in the ECM, which may be due to inhibited androgen receptor (AR) signaling, thus resulting in increased 4-*O*-sulfotransferase CHST11 expression.	[131]
Gastric cancer	Tumor tissue	CS has a 10-fold increase in 6-*O*-S and non-disaccharide units, while 4-*O*-S disaccharides were correspondingly decreased.	[132]
Ovarian adenocarcinoma	Tumor tissue	High expression of CS-E in the ECM of ovarian adenocarcinoma enhances vascular endothelial growth factor (VEGF) mediation.	[133]
Lung cancer	Lewis lung carcinoma cells	The proportion of Δ4.5HexA-GalNAc-4, 6-*O*-disulfate was higher in highly metastatic lung cancer cell lines than in low metastatic cell lines.	[134]
Breast cancer	Tumor tissue	HA was significantly increased in 143 tumor tissue samples, indicating that HA is directly involved in breast cancer metastasis.	[143]
Breast cancer	Breast tumor cells (MDA-MB-231 cells)	LMW-HA activates actin filament-associated protein (AFAP-110) to bind to F-actin, resulting in nuclear translocation of myeloiddifferentiation factor (MyD88)/NF-xB and enhanced expression of pro-inflammatory cytokines IL-1β and IL-8. AFAP-110 binding with F-actin also promoted tumor cell metastasis.	[149]
Diabetes mellitus 1 type (T1D)	Pancreas	HSPGs such as Col18 and syndecan-1 showed significant loss in T1D human islets.	[162]
Diabetes mellitus		GAG (including perlecan) sulfation patterns affected amyloid fibril formation. The order of impact was heparin > N-desulfated acetylated heparin > fully desulfated N-sulfated heparin > fully desulfated N-acetylated heparin.	[175]
Diabetic nephropathy	Glomerulus	Heparin sulfate 6-*O*-S plays an important role in extracellular matrix remodeling. Regulation of VEGFA and FGF2 signaling was achieved by increasing the expression of 6-*O*-endosulfatases Sulf1 and Sulf2 by the transcription factor Wilms’ Tumor 1 (WT1).	[183]
Diabetic nephropathy	Kidneys of rats	HS has less *N*-S in the GBM of diabetic rats compared to the normal group.	[184]
Diabetic nephropathy	Kidneys of rats	Renal CS/DS content decreased, accompanied by a decrease in the degree of sulfation, particularly 4, 6-*O*-sulfated GalN content.	[187]
Diabetic nephropathy	Renal cortex of diabetic db/db mice	4-*O*-disaccharide sulfate significantly decreased from 65% to 40%, whereas 6-*O*-S disaccharide decreased from 11% to 6% and non-sulfated disaccharide increased from 21.5% to 51% in the renal cortex of diabetic db/db mice.	[186]
T2D	Urine	The contents of total GAGs, CS/DS, and HS in urine were significantly higher than those in healthy subjects.	[188]
Diabetes mellitus	Urine	6-*O*-S and the 6-*O*-S/4-*O*-S ratio in the urine of diabetic patients with microalbuminuria were significantly increased compared with the healthy group.	[189]
Alzheimer’s disease (AD)	Brain	4-*O*-S and 6-*O*-S CS are found in neurofibrillary tangles (NFTs) of AD patients, while only 4-*O*-S CS is found in senile plaques (SPs).	[207]
Multiple sclerosis	White matter	CSPGs (versican, neurocan, and aggrecan) and DSPGs were mainly located at the edge of active plaques, while the content of CSPGs in the active center of MS plaques was significantly decreased, possibly due to the internalization of PGs in PNNs by their foam macrophages together with myelin.	[208]
AD	Brain	PGs were more abundant in areas with amyloid plaques and neurofibrillary tangles. For example, relative to healthy individuals, the total PGs increased 1.6-fold in the AD hippocampus and 3.4-fold in the superior frontal gyrus (superior gyrus frontalis). Among them, HSPGs increased the most.	[217]
AD	Occipital neocortical and hippocampal tissue	Fibrillar Aβ plaques and nonfibrillar Aβ plaques contained high levels of N-sulfated HS, while N-sulfation was very low in nonfibrillar Aβ plaques.	[218]
AD	Cerebral cortex	The critical sites for binding of heparin sulfate to β-amyloid (Aβ) fibrils contain 2-*O*-S IdoA and *N*-S from the human cerebral cortex, whereas binding of Aβ monomers requires 6-*O*-S on GlcN residues.	[219]
Parkinson’s disease (PD)	Neuronal cells	The internalization of α-syn aggregates strongly depends on the cell surface HS and is associated with their total sulfation level.	[224]
COVID-19 infection		*N*-S, 2-*O*-S, and 6-*O*-S in HS and 6-*O*-S, 2-*O*-S, 3-*O*-S, and *N*-S in HP were critical for competitive binding to SARS-CoV-2 spike protein.	[241]
COVID-19 infection		The presence of 3-*O*-S in HS contributes to the recognition and binding of SARS-CoV-2 spikes in vitro.	[242]
COVID-19 infection		3-O-sulfotransferase 3B overexpression and glycocalyx sulfation degree were too low to promote SARS-CoV-2 infection under pathological conditions.	[243]
Idiopathic pulmonary fibrosis	Lung	CS/DS, HA, and the CS/DS ratio increased significantly. CS/DS increases in 4-*O*-S, 6-*O*-S, and 2-*O*-S disaccharide units and decreases in non-sulfated disaccharides, resulting in significant increases in sulfated levels. Similarly, significant increases were observed in *N*S, 2-*O*-S, and 6-*O*-S disaccharides of HS, particularly the UA2S–GlcNS6S unit.	[247]
Asthma	Endobronchial biopsy specimens	Significant increases in PG (versican, biglycan, and decorin) content at the lesion were observed in biopsy specimens from asthma cases.	[248]
Chronic obstructive pulmonary disease	Lung	HS increased significantly, and its sulfation pattern was related to the COPD stage, while CS/DS did not change significantly. The 2-*O-*S and *N*S of HS increased during the fourth phase of COPD, while 6-*O*-S did not change.	[249]
Cystic fibrosis	Lung	GAG expression is elevated in cystic fibrosis and abnormal sulfation of GAG can be found in bronchial epithelial cells of patients.	[250,251]
Familial urolithiasis	Urine	The total sulfated GAG, CS, and HS contents in urine all decreased, while the HA content was increased, and the proportion of HS in total sulfated GAGs was increased.	[259]
Kidney stones	Urine	Black South Africans were found to have higher CS levels in urine than whites (kidney stones were significantly more prevalent in whites than blacks in South Africans), suggesting that higher CS levels may inhibit kidney stone formation.	[260]

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
