# Peer review of "The Alterations and Roles of Glycosaminoglycans in Human Diseases"

_polymers, 2022, doi:10.3390/polym14225014_

Round 1
Reviewer 1 Report
The present review about the alterations and roles of glycosaminoglycans in human diseases does not cover a gap in the literature. There are several recent reviews published in the last 2-3 years about glycosaminoglycans and diseases (this is not an exhaustive list):
Kearns, F. L., Sandoval, D. R., Casalino, L., Clausen, T. M., Rosenfeld, M. A., Spliid, C. B., Amaro, R. E., & Esko, J. D. (2022). Structural Biology Spike-heparan sulfate interactions in SARS-CoV-2 infection. Current Opinion in Structural Biology, 76, 102439.
Burgess, J. K., & Harmsen, M. C. (2022). Chronic lung diseases: entangled in extracellular matrix. European Respiratory Review, 31(163).
Ennemoser, M., Pum, A., & Kungl, A. (2021). Disease-specific glycosaminoglycan patterns in the extracellular matrix of human lung and brain. Carbohydrate Research, 108480.
Sorin, M. N., Kuhn, J., & Stasiak, A. C. (2021). Structural Insight into Non-Enveloped Virus Binding to Glycosaminoglycan Receptors: A Review. Viruses 1–11.
Faria-Ramos, I., Poças, J., Marques, C., Santos-Antunes, J., Macedo, G., Reis, C. A., & Magalhães, A. (2021). Heparan sulfate glycosaminoglycans: (un)expected allies in cancer clinical management. Biomolecules, 11(2), 1–28.
Chhabra, M., Doherty, G. G., See, N. W., Gandhi, N. S., & Ferro, V. (2021). From Cancer to COVID-19: A Perspective on Targeting Heparan Sulfate-Protein Interactions. Chemical Record, 1–16.
Additionally, several recent relevant citations of this area of study are omitted, as the reviews listed above.
This review is non-very relevant and of interest to the scientific community. In any case, I believe the review would improve significantly if all the information cited were summarized by disease in appropriate tables.
ADDITIONAL SUGGESTIONS
Figure 1 and caption:
The figure can be misleading since it is showing the maximum degree of sulfation for each GAG type. As example, chondroitin sulfate A is sulfated at carbon 4 of the GalNAc sugar, chondroitin sulfate C is sulfated at carbon 6 of the GalNAc sugar, chondroitin sulfate D is sulfated at carbon 2 of the GlcA and 6 of the GalNAc sugar, and chondroitin sulfate E is sulfated at carbons 4 and 6 of the GalNAc sugar. Therefore, I suggest adding an additional clarification in the figure caption.
Lines 75-77, “It is well known that GAGs/PGs differ in their structure and distribution in different cells/tissues. Because the biosynthesis of GAG/PGs can be affected by tissue-specific expression, activity level, and specificity of biosynthetic enzymes [22-24]”:
Rewrite the sentence, in particular the position of the conjuction “because”.
Line 190:
To specify the cancer type (breast): "Compared to nonfatal breast cancers, the sulfation degree of HS…"
Line 221, In most cancer tissues, the CS content is increased [111-115]:
References 111-115 cite studies on glycosaminoglycans/ECM characterization in some types of cancers, as breast, colorectal, lung and renal cancers. Therefore, I recommend to substitute the beginning of the phrase “in most cancer tissues” for “in some cancer tissues”.
Lines 228-230:
References 125-127 are unrelated to DS levels on tumors and are related to the regulation of apoptosis by sialic acids. Remove these references.
Lines 269-270, “It is believed that an in-depth study of GAGs in tumor cells is one of the important ways to solve the cancer problem.”:
This sentence is too general and does not add anything to the previous comments in this section.
Lines 372-373, “This may be due to 6-O-S CS altering the protective effect of PNNs and facilitating the progression of AD lesions”:
This cannot be deduced from the work cited in reference 205.
Line 404:
Indicate the reference number corresponding to Zhou et al.
Lines 404-405: The minimal binding sequence of HS is HexA–GlcNS–IdoA2S-GlcNS6S; IdoA and 6-O-S are required for binding.
The reference is missing for this sentence. Also, it is described in reference 223 that the minimal binding sequence is a hexasaccharide. Which is the correct minimal lenght?
Line 448:
Indicate the reference number corresponding to Kim et al. in the same sentence.
Line 453:
I recommend to include some missing recent (last year) references on SARS-CoV-2 and GAGs. As example, Fiona L. Kearns, Daniel R. Sandoval, Lorenzo Casalino, Thomas M. Clausen, Mia A. Rosenfeld, Charlotte B. Spliid, Rommie E. Amaro, Jeffrey D. Esko, Spike-heparan sulfate interactions in SARS-CoV-2 infection, Current Opinion in Structural Biology, Volume 76, 2022.
Line 475:
Indicate the reference number corresponding to Kim et al. in the same sentence.
Figure 3:
I believe that this figure may be misleading. As stated in the figure caption, the levels, alteration of those levels and type of GAGs depend on cancer type and tissue. I would strongly recommend to substitute this figure with summary table(s) including references and GAGs changes by pathology type.
Reviewer 2 Report
The article entitled "Alterations and Roles of Glycosaminoglycans in Human Diseases "shows a comprehensive work on the role of glycosaminoglycans in different topics. The authors describe various potential GAGs' influences on atherosclerosis, cancers, diabetes, etc. The sections on atherosclerosis and cancers if fairly presented with a sound conclusion. For the ordinary reader, it is difficult to follow all shortages in the text.
However, in diabetic nephropathy, the alteration of GAGs is found mainly in advanced disease. I suggest that the elevation of GaGs levels in urine/serum in the early stage of the disease is not so clear and should be noted as such.
The role of GAGs in neurodegenerative diseases is fairly described. As for the viral infections, The sentence "Therefore, GAGs can be used as targets for the treatment of virus infections and have great potential for application in vaccine development, treatment regimens...is highly speculative and should be rewritten or even omitted. As for "disease detection "in the same, it is quite acceptable.
As for the other diseases, I will add the role of GAGs in kidney stone diseases, recently published as a risk factor.
I suggest that the section "Abbreviations "is quite acceptable, for it will undoubtedly facilitate the reading, especially with the combination of shortages (HSPG, DSPG, etc.)
Round 2
Reviewer 1 Report
The authors have addressed all my comments and suggestions thoroughly.
Only a minor spell check is required, in paragraph (284-297), line 290; I suggest to change the verbal tense, using play instead of played.
Author Response
We appreciate the reviewer for the valuable comments. We have changed the verbal tense, using “play”, “are”, “is” instead of “played”, “were”, “was” in line 290, 290 &291, 292, respectively.
“In short, GAGs/PGs are critical regulators for cancer cell proliferation and metastasis. Although changes in GAGs/PGs vary widely in different types and stages of cancer, overall HS expression is upregulated in cancer cells relative to normal cells. As important signaling molecules on the cell surface, HS overexpression increases the communication between cancer cells and the external environment, which supports the characteristics of cancer cells that are prone to proliferation and metastasis. During this period, N-S, 2-O-S, 6-O-S of HS play an important regulatory role. CS/CSPGs are also upregulated in some tumors, and at the same time, they are accompanied by increased 6-O-S levels and decreased 4-O-S levels. DS/DSPG is detected in some tumors, which suggests that DS may have antitumor activity. LMW-HA increases in many types of cancer and promotes tumor cell adhesion and migration. Thus, GAGs/PGs play a crucial role in tumor cell activity. In addition, GAGs perform their functions by interacting with proteins in the body. Therefore, the development of some GAGs analogues to inhibit their interactions may become another approach for cancer treatment”. (Lines 284-297).